# Pressurized Intraperitoneal Aerosol Chemotherapy for Peritoneal Carcinomatosis in Colorectal Cancer Patients: A Systematic Review of the Evidence

**DOI:** 10.3390/cancers16213661

**Published:** 2024-10-30

**Authors:** Marwan-Julien Sleiman, Annamaria Jelip, Nicolas Buchs, Christian Toso, Emilie Liot, Thibaud Koessler, Jeremy Meyer, Guillaume Meurette, Frederic Ris

**Affiliations:** 1Division of Digestive Surgery, University Hospitals of Geneva, 1205 Geneva, Switzerland; annamaria.jelip@hug.ch (A.J.); christian.toso@hug.ch (C.T.); emilie.liot@hug.ch (E.L.); jeremy.meyer@hug.ch (J.M.); guillaume.meurette@hug.ch (G.M.); frederic.ris@hug.ch (F.R.); 2Division of Digestive Surgery, Hôpital La Tour, 1217 Meyrin, Switzerland; nicolas.buchs@hug.ch; 3Division of Oncology, University Hospitals of Geneva, 1205 Geneva, Switzerland; thibaud.koessler@hug.ch

**Keywords:** colorectal cancer, peritoneal carcinomatosis, PIPAC, ePIPAC

## Abstract

This article stems from my research project on the role of Pressurized Intraperitoneal Aerosol Chemotherapy (PIPAC) in the treatment of peritoneal carcinoma originating from colorectal cancer. Our aim was to assess the existing literature regarding the impact, benefits, and risks associated with PIPAC for patients. The results are encouraging and suggest that PIPAC is a safe option for patients. However, monitoring the progression of peritoneal carcinomatosis and evaluating the effects of PIPAC can be challenging. These findings reinforce the need for a randomized study to better define the role of PIPAC in treating colorectal peritoneal carcinomatosis. By clarifying the implications of this innovative therapy, this research aims to provide valuable insights for clinicians, patients, and policymakers, ultimately enhancing patient care and treatment outcomes.

## 1. Introduction

Colorectal cancer is the third most common cancer in men and the second most common cancer in women [1]. In the USA,153’020 incidental cases of colorectal cancer have been diagnosed in 2023 [2]. Peritoneal carcinomatosis (PC) is found in approximately 5% of patients at the time of diagnosis of colorectal cancer and appears in 5% of patients as metachronous distant disease [3,4].

Risk factors for developing PC are locally advanced disease (T4 stage), tumor perforation, mucinous and signet ring cell histologies, positive nodal stage, right-sided tumor location and surgical resection with positive margins (R1 or R2) [5]. PC can be symptomatic with bowel obstruction or ascites and therefore associated with decreased quality of life [3]. Untreated, PC is associated with poor overall survival as low as 5.2 months [6]. With systemic chemotherapy, the overall survival does not reach more than a median of 16.3 months [7], with progression-free survival (PFS) from 5.7 to 5.8 months [8,9] due to a weak penetration of chemotherapy into the peritoneum due to limited blood flow, interstitial fibrosis and the plasma–peritoneal barrier [7].

Therefore, cytoreductive surgery (CRS) and hyperthermic intraperitoneal chemotherapy (HIPEC) were developed to overcome this limitation. The added value of CRS and HIPEC was first studied by Cashin et al. who showed an improved 2-year overall survival of 54% in the CRS+HIPEC arm when compared to 24% in the control arm with systemic chemotherapy alone [10]. This was subsequently confirmed by Verwall et al., who demonstrated improved overall survival in the CRS + HIPEC group when compared to systemic chemotherapy (22.3 months versus 12.3 months) [11,12]. To determine if the observed benefits were the effect of CRS or HIPEC, Elias et al. compared CRS+HIPEC to CRS associated with systemic chemotherapy and demonstrated no survival benefit in adding HIPEC [13]. Finally, in the PRODIGE 7 trial, Quenet et al. confirmed these latest results by using oxaliplatin-based HIPEC [14]. Furthermore, HIPEC is associated with a high morbidity (33%) and a high mortality (2.8%) [15].

Pressurized intraperitoneal aerosol chemotherapy (PIPAC) has been developed as an alternative to HIPEC. While HIPEC mediates its effect through heat, PIPAC enhances the effect of chemotherapy through pressure [16]. The advantage of PIPAC, compared to HIPEC, is that PIPAC is feasible in most patients and could be repeated. The feasibility of PIPAC includes abdominal access by laparoscopy [17]. As PIPAC is usually repeated several times, it offers the opportunity to assess response to treatment and sample remaining disease [18].

In order to assess the safety and efficacy of PIPAC in peritoneal carcinomatosis from colorectal primary cancer, we conducted a systematic literature review.

## 2. Methods

A systematic review was performed in accordance with the 2020 PRISMA (Preferred Reporting Items for Systemic Reviews and Meta-analyses) guidelines [19]. MEDLINE (through Pubmed) and the Cochrane Central Register of Controlled Trials (CENTRAL) were searched using combinations of terms including “Peritoneal carcinomatosis”, “Peritoneal metastasis”, “PIPAC”, “Pressurized intraperitoneal aerosol chemotherapy” and “Colorectal cancer”. The search strategy is summarized in Table 1. Articles were initially screened based on titles and abstracts. Original studies, in English, including patients treated with PIPAC for colorectal peritoneal carcinomatosis, were considered eligible. Case reports, non-English articles and secondary analyses were excluded. The literature search was performed up to August 2024. Two authors (MJS and JA) performed the literature screening. Eligible publications were explored after their full texts had been obtained. Any discrepancies were resolved by consensus with a third author (JM). 

The following variables were extracted from included studies and reported in tables: author, year of publication, country of the first author, study design, study population, primary tumor location, synchronous or metachronous peritoneal metastases, number of included patients, number of colorectal patients, study duration, primary outcome, chemotherapy agent used for PIPAC, systemic chemotherapy and agents, concomitant systemic chemotherapy while PIPAC, adverse effects, evolution of PCI and evolution of median OS. 

## 3. Results

### 3.1. Inclusion Process

Three-hundred and eighty-five publications were identified via database search. Three-hundred and fifty-eight publications underwent additional scrutiny after duplicates were removed. Title and abstract screening resulted in the exclusion of 342 articles. Sixteen publications underwent full-text screening. This resulted in the exclusion of 5 articles, leaving 11 publications for inclusion in the qualitative analysis (Figure 1). Of the five excluded articles, three were research protocols, one was a systematic review and one was a research letter.

### 3.2. Characteristics of Included Studies

Table 2 summarizes the main characteristics of the included studies. Five studies were retrospective cohorts [20,21,22,23,24], five were randomized controlled trials [25,26,27,28,29] and one was a prospective cohort [30]. Ten studies were European [20,21,22,23,24,25,26,27,28,30] and one study was from the USA [29]. The number of patients included ranged from 5 to 342 patients, with a total of 587 patients treated with PIPAC. All studies included patients with unresectable PC from colorectal cancer [20,21,22,23,24,25,26,27,28,29,30]. Four studies added appendiceal cancer to their populations [25,26,27,29]. Five studies documented whether PC was synchronous or metachronous [22,24,25,26,27] and six studies did not provide any information regarding the temporality of PC [20,21,23,28,29,30]. Synchronous PC was present in 75% of the population in three studies [25,26,27]. Four studies excluded patients with distant metastases other than PC [21,24,26,27], while three studies included distant metastases other than PC [22,23,28]. This information was not available for four studies [20,25,29,30].

### 3.3. Feasibility of PIPAC

The feasibility and tolerability of PIPAC were described as primary outcomes in three studies [22,27,29]. Raouf et al. described no technical failures in completing the laparoscopies, with 58% of patients completing two or more PIPAC sessions and 50% completing at least three PIPAC sessions [29]. In the study by Ellebaek et al., 79% of patients received two PIPAC sessions, 62% had three PIPAC sessions and 4% had seven PIPAC sessions [23]. Rovers et al. showed that 20% of patients had one PIPAC session, 15% had two PIPAC sessions, 35% had three PIPAC sessions, 15% had four PIPAC sessions, 10% had five PIPAC sessions and 5% had six PIPAC sessions [26]. Demtröder et al. showed that 82% of patients received several sessions of PIPAC with a mean number of 2.8 PIPAC sessions per patient [21]. Taibi et al. reported a median number of 2 PIPAC sessions per patient in the PIPAC-Ox group and 2.5 in the PIPAC-Ox + 5-FU/L group [20]. Lurvink et al. found an average of three PIPAC sessions per patient [25].

In terms of non-access rate, Tabchouri et al. reported a primary non-access rate for the first sessions of the PIPAC-Ox cycle of 21.6%. Secondary non-access for a second or third PIPAC-ox session was observed in 13.7% of patients [22]. Demtröder et al. had access to the abdominal cavity in all cases [21].

### 3.4. Chemotherapeutic Agents Administered for PIPAC

Oxaliplatin at 92 mg/m^2^ for 30 min was used as the single chemotherapy agent in PIPAC in all studies [20,21,22,23,25,26,27,28,29,30]. No adverse event reactions were described [20,21,22,23,25,26,27,28,29,30]. PIPAC-based oxaliplatin (PIPAC-OX) was first developed by Demtröder et al. with 17 patients [21]. A per-operative systemic chemotherapy agent (5FU +leucovorin) was used in five studies [20,25,26,27,29]. Taibi et al. showed that the adjunction of intravenous 5 FU + Leucovorin (PIPAC-OX + 5 FU/L) appears to have similar safety and feasibility to PIPAC-OX [20]. This result was further confirmed by Raoof et al. [29].

### 3.5. Systemic Chemotherapeutic Treatment

All studies included patients with previous systemic chemotherapeutic treatment including several lines of systemic chemotherapy [20,21,22,23,24,25,26,27,28,29,30]. Concomitant systemic chemotherapy treatment with PIPAC was part of the protocol in five studies [20,21,22,23,28]. All studies showed no difference in the safety and tolerance of PIPAC-OX associated with concomitant systemic chemotherapy [20,21,22,23,28].

### 3.6. Adverse Events and Safety of PIPAC

Adverse events after PIPAC procedure were described in six studies [20,21,22,26,28,29]. Using the common terminology criteria for adverse events (CTCAE), in the PIPAC-OX + 5 FU/L group, Taibi et al. showed 43.3% of grade 2 or higher [20]. Rovers et al. reported 15% of CTCAE grade 3 or higher [26]. Raoof et al. reported only grade 1 to 3 adverse effects of CTCAE [29]. In comparison, in the PIPAC-OX group, Taibi et al. showed 47.5% of grade 2 or higher [20]. Demtröder et al. observed CTCAE grade 3 symptoms in 23% of patients and no CTCAE grade 4 symptoms [21]. Tabchouri et al. reported 3.8% of grade 3 or higher CTCAE [22]. Kyle et al. confirmed the same results with 100% of grade 1 or 2, 40% of grade 3 and 0% of grade 4 or 5 [28].

In the two studies comparing both groups, adverse events had no significant differences between the two groups [20,29]. The most common toxicities were gastrointestinal with abdominal pain, nausea, vomiting, constipation, abdominal distention and diarrhea in both groups [20,21,22,26,28,29].

According to the Clavien–Dindo classification, grade 3 or higher post-operative complications occurred in the 10% of patients representing 3% of the procedure in the study by Rovers et al. [26]. Demtröder et al., Raoof et al. and Kile et al. noted no intraoperative complications [21,28,29]. Tabchouri et al. showed that redo surgery was necessary in 2% with a procedure-related mortality of 0.5% [22]. Ellebaek et al. recorded most of the post-operative adverse events were recorded as mild to moderate with no intraoperative complications. Only one reoperation was necessary due to iatrogenic perforation of the jejunum on day one post-surgery [23].

### 3.7. Peritoneal Carcinomatosis Index

Initial median PCI, previous to any PIPAC treatment, was reported in 7 studies [20,21,23,24,26,28,29] as seen in Table 3. Median PCI ranged from 10.7 to 31 [20,21,23,24,26,28,29]. Three studies reported a decrease in the average PCI after repeated sessions of PIPAC [20,21,26]. Rovers et al. had a median PCI of 30 at baseline and 25 post-treatment [26]. The two other studies did not quantify the decrease in PCI [20,21]. PCI did not decrease in three studies [22,28,29]. Downstaging of the PCI allowed for performing CRS and HIPEC in three studies [21,24,29].

### 3.8. Survival Outcomes

Overall survival was reported in 8 studies and ranged from 8 to 37.8 months [20,21,22,23,24,26,28,29]. In the PIPAC-OX + 5 FU/L group, Rovers et al. described an OS of 8 months [26]. Raouf et al. confirmed a median OS of 12 months in the PIPAC trial [29]. Taibi et al. showed that a median OS was 13 months [20].

In comparison, in the PIPAX-OX group, OS was 17 months [20]. Ellebaek et al. recorded a median survival rate of 20.5 months following the start of the first PIPAC session [23]. Hubner et al. found a median OS from the first PIPAC of 9.4 months [24]. Kyle et al. reported a median OS of 11.6 months from the first PIPAC [28].

Only one study of randomized PIPAC-OX + 5 FU/L vs. PIPAX-OX with no significant differences in OS between the two groups [20].

Tabchouri et al. showed a median OS of 13 months with a variation in the function of the number of PIPAC procedures. Patients who underwent one or two PIPAC-OX cycles had a median OS of 8.8 months compared with those who underwent three or more cycles, who had a median OS of 17.2 months [22].

Only three studies could compare OS in the PIPAC-OX group with patients not receiving PIPAC from other studies [22,23,29]. They all found a better OS in the PIPAC-OX group but, due to the limitations of each study, no conclusions could be determined.

### 3.9. Quality of Life

Quality of life (QOL) was reported in four studies representing 147 patients [22,25,28,30] by using three questionnaires: the EuroQol EQ-5D-5L, EORTC QLQ-C 29 and EORTC QLQ-CR30 as seen in Table 4.

Tabchouri et al. recorded that QOL indicators were stable between PIPAC-OX cycles with a small but not statistically significant trend of improvement of most functional scales [22]. Kyle et al. observed that no statistical assessment can be performed but the patient’s global health status did not show any deterioration following PIPAC. Similarly, physical, emotional and cognitive functioning scales showed no clear trend in any direction [28]. Lurvink et al. showed that the majority of patient-reported outcomes (PRO) worsened 1 week after the first procedure and returned to baseline at all subsequent time points [25]. 

## 4. Discussion

We conducted a systematic literature review to assess the safety and efficacy of PIPAC in the treatment of patients with unresectable colorectal PC.

The feasibility of PIPAC has been demonstrated since 2015 by Demtröder et al. [21]. This result has now been confirmed by all retrospective and RCT studies included in this review. In 2021, Tabchouri et al. showed the primary non-access rate for the first PIPAC-Ox cycle was 21.6% [22]. More recently, Raouf et al. reported evidence from phase I of the PIPAC trial in the USA, where no technical failure was reported. This demonstrates the feasibility of the PIPAC procedure and of repeatedly obtaining laparoscopic access [29]. Furthermore, formal training, adherence to the standardized procedure and established safety protocol allow for the safe implementation of PIPAC with no learning curve [31]. On the contrary, CRS and HIPEC can usually be performed only once due to the high risk of surgical complications [10,11,12].

For colorectal PC, two regimens of intraperitoneal (IP) agents of PIPAC were usually described as cisplatin with doxorubicin or oxaliplatin. Oxaliplatin is more routinely administered currently, and eleven studies assess oxaliplatin exclusively. Oxaliplatin was first used with fear due to toxicity, which was encountered after IP administration in the form of HIPEC [32]. PIPAC-Ox was first developed by Demtröder et al. with the largest study on 17 patients with colorectal PC. PIPAC-OX was then further used in all different studies and showed a majority of grade 1 toxicity confirming the tolerability of PIPAX-OX [20,29]. By comparison, Sgarbura et al. included 101 patients with 251 PIPAC-OX procedures from gastrointestinal tract cancer. A total of 65.4% of patients had colorectal primary cancer and no major toxicity was described in the global population [33]. On the contrary, the HIPEC hematological toxicity rate was lower; present in 18% of patients in the PROPHYLOCHIP-PRODIGE 15 study [34] and leading to 28% of patients discontinuing treatment early, due to toxicity, in PRODIGE 7 [14]. This result was not encountered in the PIPAC trial.

In this review, all retrospective cohorts and RCT selected used PIPAC-OX at the same dosage of 92 mg/m^2^ for 30 min. Dumont et al. observed two dose-limiting toxicities of systemic therapy with repetitive PIPAC-OX at 140 mg/m^2^. The investigators defined a maximum tolerated dose (MTD) of 92 mg/m^2^ for 30 min [35]. This dosage of PIPAC-OX is low, compared to the one used in HIPEC (200–460 mg/m^2^) but similar to the IV dose of oxaliplatin (85 mg/m^2^) [14]. This was explained for two reasons: PIPAC-OX is repeated and so gives a median cumulated oxaliplatin dose of 697 mg, which is higher than HIPEC [36]. Then, higher plasma oxaliplatin concentrations were observed after PIPAC. These plasma concentrations are slightly lower than with high oxaliplatin doses (260 mg/m^2^) used in HIPEC [37]. Lurvink et al. reported similar high plasma concentrations using electrostatic PIPAC with oxaliplatin (92 mg/m^2^) [27]. Finally, Dumont et al. suggested the feasibility and low toxicity rate of PIPAC-OX with intraoperative intra-venous 5 fluorouracil and leucovorin in patients with unresectable colorectal PC [35]. Taibi et al. also confirm that the safety and feasibility of PIPAX-OX + 5 FU/L appears to be similar to the safety and feasibility of PIPAC-OX alone [20].

All the studies suggested that PIPAC is usually safe and well tolerated. A majority of grade 2 or lower CTCAE adverse effects with no intraoperative complication have been described [20,21,22,23,26,28,29]. The interpretation of these results is subject to debate. The adverse effects reports were not stratified for PIPAC-OX monotherapy and for PIPAC-OX with concomitant systemic therapy. A systemic review by Lurvink et al. found a rate of adverse effect of 19% and already evocated this stratification issue [38]. Nevertheless, these results are still better than HIPEC, which is associated with high morbidity (33%) and high mortality (2.8%) [15], compared to 59.4% of grade 3 events in the systemic chemotherapy group [39].

Initially, PCI has been described as an independent prognostic factor for PC [40]. The evolution of response using laparoscopies PCI was not correlated with OS [29]. Indeed, we observed that PCI is a suboptimal tool for the evaluation of the response to PIPAC, as it is difficult to differentiate macroscopic progression and treatment-induced fibrosis. This is a limitation not only specific to PIPAC. In the PROPHYLOCHIP-PRODIGUE 15 study, the macroscopic diagnosis of peritoneal recurrence was not histologically confirmed in 31% of patients [34]. Furthermore, repeated PIPAC-OX appeared to develop significant adhesion formation limiting visual PCI assessment. The peritoneal regression grading score (PRGS) has been proposed to evaluate response after locoregional peritoneal therapies and grade the histological appearance of tumor biopsies from 1 to 4. PRGS 1 is a complete response with no tumor cell and PRGS 4 is no response with a tumor cell present with any regression. In fact, Raouf et al. demonstrated no correlation between response by PCI or mean PGRS with OS. OS was correlated by imaging RECIST criteria and stable disease [29]. This result was confirmed by Hübner et al., who showed that PRGS and PCI were no independent prognostic values [24]. For comparison, De Simone et al. evaluated the disease control rate (DCR), the proportion of patients achieving complete radiological response, a partial response and stable disease divided by the total number of patients enrolled, according to the RECIST criteria after two cycles of PIPAC. They reported a DCR of 35% and the median OS of the PIPAC population was 18.1 months without any statistically significant differences due to systemic chemotherapy [41]. Currently, RECIST criteria are suggested as potential surrogate markers for survival [29]. Another limitation of PRGS was that it can be challenging to make a distinction between intermediate categories of PRGS 2 and 3.

In 2015, Odendahl et al. showed stabilization of QOL under PIPAC with a functional score remaining stable and a non-deterioration of gastrointestinal symptoms [42]. Lurvink et al. showed that the majority of PRO worsened 1 week after the first procedure and returned to baseline at all subsequent time points [25].

In comparison, Cashin et al. and Andre et al. both reported a worsening of several PROs during treatment with systemic chemotherapy [43,44]. After cytoreductive surgery and HIPEC, Cashin et al. demonstrated a deterioration of several PROs [43]. A second study reported that several PROs deteriorated shortly after surgery but recovered rapidly and remained stable during follow-up [45]. No studies comparing PROs during PIPAC and systemic chemotherapy or HIPEC were found. Only Van de Vlasakkler et al. compared PROs between PIPAC and primary tumoral surgery. It showed that PIPAC-OX resulted in significantly worse abdominal pain than primary tumor surgery. This is mainly explained due to chemical peritonitis after the PIPAC procedure. A better control of abdominal pain after the PIPAC procedure is then recommended [30].

The limitations observed across these 14 studies are significant and must be carefully considered when interpreting their findings. One major limitation is the relatively small sample size of colorectal cancer patients, which reduces the statistical power and may limit the generalizability of the results to a broader population. On the other hand, there is considerable heterogeneity in the treatment regimens employed, with some studies using PIPAC-OX as a monotherapy while others combine it with systemic chemotherapy. This variation complicates direct comparisons between studies and makes it challenging to draw definitive conclusions about the efficacy of PIPAC-OX.

Another important limitation is the inconsistent reporting of outcomes, with some studies failing to stratify results based on treatment regimens, which hinders a clear understanding of the specific benefits or drawbacks of PIPAC-OX when combined with systemic therapy versus when used alone. Beyond these, other potential biases that may affect these studies include selection bias, where the patient populations may not fully represent the broader colorectal cancer population, and publication bias, where studies with positive or more favorable results are more likely to be published, while negative or inconclusive findings may be underreported. To complete, performance bias is also a concern, as differences in how treatments are administered or monitored across centers can influence outcomes. Detection bias, where variations in how outcomes are measured or assessed, may lead to inconsistencies in evaluating the true effectiveness of PIPAC-OX.

These limitations and potential biases highlight the need for more standardized, larger-scale randomized studies with clear, uniform protocols to provide more reliable and comparable data, ultimately improving the evidence base for PIPAC-OX in colorectal peritoneal carcinomatosis.

To the best of our knowledge, there is no phase III prospective RCT comparing the effect of systemic chemotherapy alone versus PIPAX-OX and systemic chemotherapy for unresectable colorectal PC. On a personal side of view, we recognize an urgent need for this trial is necessary to confirm the effectiveness of PIPAC-OX. Indeed, at the molecular level, PC is very similar to the primary tumor, contributing to the idea that the problem is not drug sensitivity but drug distribution. There is a clear unmet medical need in the field of metastatic CRC with PC, which we want to address with this trial proposal.

This literature review was not registered, and no protocol was prepared. No financial support was received, and all authors declare no conflicts of interest. All data can be found in Table 1 or Table 2 as presented in this study.

## 5. Conclusions

This systematic review of the literature confirmed that PIPAC is feasible, safe and tolerable. Many questions remained open, the first and most important is the true impact of PIPAC on overall survival, which needs to be explored by a randomized clinical trial. Other aspects such as imagery assessment of response, pathology markers for surrogacy of survival and multi-drug testing should also be investigated.

## Figures and Tables

**Figure 1 cancers-16-03661-f001:**
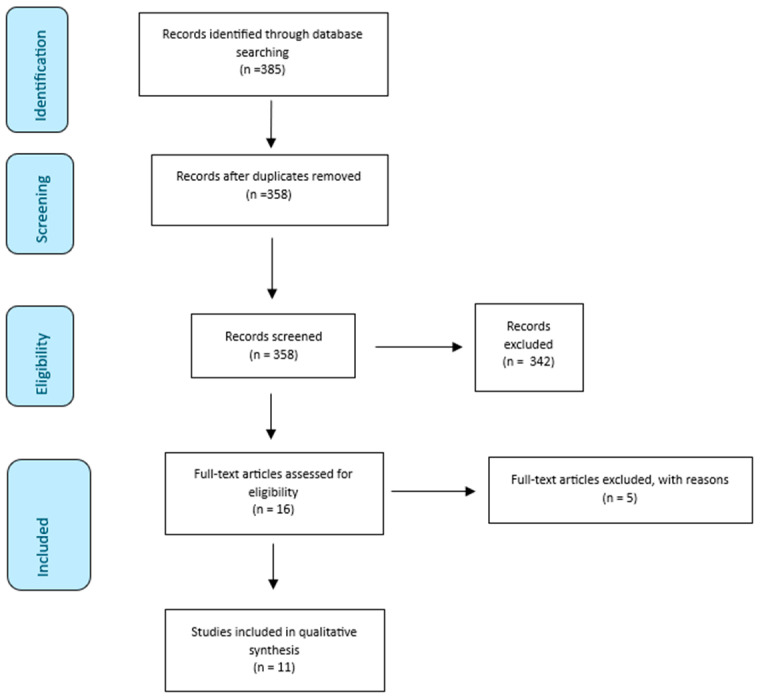
Prisma 2020 flow diagram.

**Table 1 cancers-16-03661-t001:** Literature search strategy.

	Literature Search Strategy	
	Search Terms	Occurrence
Medline	(ROD[All fields] AND ((‘Peritoneal carcinomatosis’ [MeSH Terms] OR Peritoneal Metastasis)) AND ((PIPAC) OR (Pressurized intraperitoneal aerosol chemotherapy)) AND ((Colorectal cancer))	82
Cochrane		303

**Table 2 cancers-16-03661-t002:** Main characteristics of the studies with colorectal cancer patients receiving all PIPAC-OX. NA: Not available.

Studies	Years	Countries	Authors	Study Design	Study Population	Primary Tumor	Synchronous PC	Metachronous PC
A.Taibi et al., Ann Surg Oncol [20]	2022	France	A.Taibi	RC	Unresectable and isolated cPM	CRC	NA	NA
RJ Lurvink et al., Surgical endoscopy [25]	2021	Netherland	Lurvink RJ	RCT	Isolated unresectable PM	CRC or appendiceal carcinoma	75%	25%
K.P Rovers et al., Ann Surg Onco [26]	2021	Netherland	K P. Rovers	RCT	Unresectable PM	CRC or appendiceal carcinoma	75%	25%
C. Demtröder et al., Colorectal Disease [21]	2015	Germany	C. Demtröder	RC	Advanced therapy resistant PM	CRC	NA	NA
N. Tabchouri et al., Ann Surg Oncol [22]	2021	France	N.Tabchouri	RC	PM with progression under SC	CRC	44%	56%
R.L Lurvink et al., Ann Surg Oncol [27]	2020	Netherland	R J. Lurvink	RCT	Unresectable PM	CRC or appendiceal carcinoma	75%	25%
SB Ellebaek, Pleura and Peritoneum [23]	2020	Denmark	SB Ellebaek	RC	Unresectable PM	CRC	NA	NA
M.Hubner et al., Annals of surgery [24]	2022	Switzerland	M.Hubner	RC	CRC PM	CRC	61%	39%
P.Kyle et al., Pleura and Peritoneum [28]	2023	UK	P.Kyle	RCT	CRC PM with no CRS +/− HIPEC	CRC	NA	NA
M.Raoof, Ann Surg Oncol [29]	2023	USA	M.Raoof	RCT	Appendiceal or CRC PM	CRC or appendiceal carcinoma	NA	NA
Van de Vlasakker, Scientific reports [30]	2023	Netherland	Van de Vlasakker	Prospective	CRC	CRC	NA	NA

**Table 3 cancers-16-03661-t003:** Results from the included studies with colorectal cancer patients receiving all PIPAC-OX. NA: Not available.

Studies	Nbr of Patients	Study Duration	Primary Outcome	Previous SC	PCI	CRS
A.Taibi et al., Ann Surg Oncol [20]	131	Jan 2015–April 2020	OS and PFS	Yes	18	NA
RJ Lurvink et al., Surgical endoscopy [25]	20	Oct 2017–Sept 2018	PROS	Yes	NA	NA
K.P Rovers et al, Ann Surg Onco [26]	20	Oct 2017–Sept 2018	Adverse events	Yes	31	NA
C. Demtröder et al., Colorectal Disease [21]	17	Oct 2012–Feb 2014	Pathological response, Adverse effect	Yes	16	No
N. Tabchouri et al., Ann Surg Oncol [22]	102	April 2014–June 2018	Tumor response/adverse effect	Yes	NA	Yes
R.L Lurvink et al., Ann Surg Oncol [27]	20	Oct 2017– April 2019	Feasibility, Safety, Efficacy	Yes	NA	NA
SB Ellebaek, Pleura and Peritoneum [23]	24	Oct 2015–Feb 2019	Objective tumors responses	Yes	10.7	No
M.Hubner et al., Annals of surgery [24]	256	NA	OS	Yes	18	Yes
P.Kyle et al., Pleura and Peritoneum [28]	5	Jan 2019–Jan 2022	PFS	Yes	25.7	No
M.Raoof, Ann Surg Oncol [29]	12	Aug 2020–Jan 2023	Safety	Yes	28	Yes
Van de Vlasakker, Scientific reports [30]	342	Oct 2015–Febr 2019	PROS	NA	NA	NA

**Table 4 cancers-16-03661-t004:** Further Results from the included studies with colorectal cancer patients receiving all PIPAC-OX. NA: Not available.

Studies	OS	Adverse Effects (>Grade 3)	Quality of Life
A.Taibi et al., Ann Surg Oncol [20]	13 to 17	9–10%	No
RJ Lurvink et al., Surgical endoscopy [25]	NA	NA	yes
K.P Rovers et al., Ann Surg Onco [26]	8	8%	No
C. Demtröder et al., Colorectal Disease [21]	15.7	23%	No
N. Tabchouri et al., Ann Surg Oncol [22]	13	3.80%	yes
R.L Lurvink et al., Ann Surg Oncol [27]	NA	NA	No
SB Ellebaek, Pleura and Peritoneum [23]	37.6	3	No
M.Hubner et al., Annals of surgery [24]	11.9	NA	No
P.Kyle et al., Pleura and Peritoneum [28]	11.6	30%	yes
M.Raoof, Ann Surg Oncol [29]	12	0%	No
Van de Vlasakker, Scientific reports [30]	NA	NA	yes

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
