# Peer review of "Pressurized Intraperitoneal Aerosol Chemotherapy for Peritoneal Carcinomatosis in Colorectal Cancer Patients: A Systematic Review of the Evidence"

_cancers, 2024, doi:10.3390/cancers16213661_

Round 1

Reviewer 1 Report

Comments and Suggestions for Authors

The manuscript is a systematic review on PIPAC for treating peritoneal carcinomatosis in colorectal cancer patients. They found that PIPAC is generally well-tolerated with mild to moderate adverse effects and shows some potential for stabilizing quality of life. 

Overall, the manuscript is well written, and addresses and important topic. some suggestions:

- why were studies in French specifically considered? I would have expected to include only studies published in English. please clarify

- please provide more information on the methods for assessing study quality or risk of bias

- While the discussion touches on the need for randomized trials, there could be a clearer acknowledgment of the limitations of the systematic review itself, such as potential publication bias, heterogeneity in the included studies, or lack of long-term follow-up data.

- Although the search strategy appears thorough, it’s important to verify if all the latest studies are included, since the research was concluded in January 2024.

- Although the manuscript calls for randomized trials, it could go further by suggesting specific research areas.

- please also expand on Authors' personal views.

Author Response

Comment 1 : why were studies in French specifically considered? I would have expected to include only studies published in English. please clarify

Answer 1 : Thank you very much for your comment. We have included only English studies and modified the manuscript to make it clearer.

Comment 2 : please provide more information on the methods for assessing study quality or risk of bias

Answer 2 : Thank you very much for your comment. We agree with your observation. In the manuscript, we have expanded the discussion on study quality and risk of bias.

Comment 3 : While the discussion touches on the need for randomized trials, there could be a clearer acknowledgment of the limitations of the systematic review itself, such as potential publication bias, heterogeneity in the included studies, or lack of long-term follow-up data.

Answer 3 : Thank you very much for your comment. Indeed, we agree with your observation. In the manuscript, we acknowledge the limitations of the review itself in the discussion.

Comment 4 : Although the search strategy appears thorough, it’s important to verify if all the latest studies are included, since the research was concluded in January 2024.

Answer 4 : Thank you for this comment. We have updated the research until August 2024 and corrected the article in September 2024. I should have updated it in the manuscript. We apologize for this oversight and have made the corrections in the manuscript. Until August 2024, no new articles were published.

Comment 5:  Although the manuscript calls for randomized trials, it could go further by suggesting specific research areas.

Answer 5: Thank you for your comment. We have developed and proposed a randomized trial at the end of the manuscript.

Comment 6 :  please also expand on Authors' personal views

Answer 6: Thank you for your comment. We have expanded on the authors' personal views at the end of the manuscript.

Reviewer 2 Report

Comments and Suggestions for Authors

This paper describes a systematic review about pressurized intraperitoneal aerosol chemotherapy for peritoneal carcinomatosis in colorectal cancer patients. This a useful review about that subject. The following revisions should made:

- The authors excluded a systematic review from the list of paper to be considered. All the reviews about this subject should be discussed here to support the novelty of the present revision.

- The format of the citations within the text are not common and should be modified, for example "/" should be replaced by ",", ...

- The columns of the tables that are constant values should be deleted and the values described in the tables captions. The captions of the tables should be on the top of the table - for example, in Table 3 the "Nbr. of CR patients" is always 100%, and the "Chemotherapy agent used for PIPAC" is always the same. 

Author Response

Comment 1 : The authors excluded a systematic review from the list of paper to be considered. All the reviews about this subject should be discussed here to support the novelty of the present revision

Answer 1 : Thank you for your comment. We have excluded all secondary publications from our search. However, we have included the systematic review in the manuscript discussion.

Comment 2 : The format of the citations within the text are not common and should be modified, for example "/" should be replaced by ",", ..

Answer 2 : Thank you for your comment. We have changed the format of citations: '/' has been replaced with ','.

Comment 3 : The columns of the tables that are constant values should be deleted and the values described in the tables captions. The captions of the tables should be on the top of the table - for example, in Table 3 the "Nbr. of CR patients" is always 100%, and the "Chemotherapy agent used for PIPAC" is always the same. 

Answer 3 : Thank you for your comment. We have changed the table as requested.